# Is the Validity of Logistic Regression Models Developed with a National Hospital Database Inferior to Models Developed from Clinical Databases to Analyze Surgical Lung Cancers?

**DOI:** 10.3390/cancers16040734

**Published:** 2024-02-09

**Authors:** Alain Bernard, Jonathan Cottenet, Catherine Quantin

**Affiliations:** 1Department of Thoracic and Cardiovascular Surgery, Dijon University Hospital, 21000 Dijon, France; alain.bernard@chu-dijon.fr; 2Service de Biostatistiques et d’Information Médicale (DIM), CHU Dijon Bourgogne, Inserm, Université de Bourgogne, CIC 1432, Module Épidémiologie Clinique, 21000 Dijon, France; jonathan.cottenet@chu-dijon.fr; 3CESP, Inserm, UVSQ, Université Paris-Saclay, 94807 Villejuif, France

**Keywords:** model performance, hospital database, clinical database, brier score, area under the receiver operating characteristic, discrimination, calibration

## Abstract

**Simple Summary:**

In national hospital databases, certain prognostic factors cannot be taken into account. Our objective was to estimate the performance of two models based on the Epithor clinical database and the French hospital database. The performance of the models was assessed with the Brier score, the area under the receiver operating characteristic (AUC ROC) curve, and the calibration of the model. For the Epithor and hospital databases, the training dataset (70% of the initial data) included 10,516 patients (with, respectively, 227 (2.16%) and 283 (2.7%) deaths) and the validation dataset (30% of the initial data) included 4507 patients (with, respectively, 93 (2%) and 119 (2.64%) deaths). The Brier score values were similar in the models of the two databases. For validation data, the AUC ROC curve was 0.73 [0.68–0.78] for Epithor and 0.8 [0.76–0.84] for the hospital database. The slope of the calibration plot was less than 1 for the two databases. This work showed that the performance of a model developed from a national hospital database is nearly as good as a performance obtained with Epithor, but it lacks crucial clinical variables.

**Abstract:**

In national hospital databases, certain prognostic factors cannot be taken into account. The main objective was to estimate the performance of two models based on two databases: the Epithor clinical database and the French hospital database. For each of the two databases, we randomly sampled a training dataset with 70% of the data and a validation dataset with 30%. The performance of the models was assessed with the Brier score, the area under the receiver operating characteristic (AUC ROC) curve and the calibration of the model. For Epithor and the hospital database, the training dataset included 10,516 patients (with resp. 227 (2.16%) and 283 (2.7%) deaths) and the validation dataset included 4507 patients (with resp. 93 (2%) and 119 (2.64%) deaths). A total of 15 predictors were selected in the models (including FEV1, body mass index, ASA score and TNM stage for Epithor). The Brier score values were similar in the models of the two databases. For validation data, the AUC ROC curve was 0.73 [0.68–0.78] for Epithor and 0.8 [0.76–0.84] for the hospital database. The slope of the calibration plot was less than 1 for the two databases. This work showed that the performance of a model developed from a national hospital database is nearly as good as a performance obtained with Epithor, but it lacks crucial clinical variables such as FEV1, ASA score, or TNM stage.

## 1. Introduction

Lung cancer is one of the deadliest cancers worldwide despite the therapeutic progress made in recent years [1,2]. In France, it is the third most common cancer [3] and remains the leading cause of cancer mortality in men and the second in women, causing 22,761 and 10,356 deaths in 2018, respectively. Age-standardized all-stage net survival at 5 years is 17% (16% in men, 20% in women), and 10% at 10 years (9% in men, 13% in women). For surgically treatable early stages, 5-year survival ranges from 77 to 92% for stage IA, 68% for stage IB, 60% for stage IIA, 53% for stage IIB [4].

For early stage bronchial cancers (stages IA and IB), lung resection surgery associated with mediastinal lymph node dissection is the first-line treatment [5,6]. Thus, for patients with normal respiratory capacity, the standard treatment will be lung lobectomy combined with a mediastinal lymph node dissection. In some cases, for patients with a tumor ≤2 cm in diameter, in the absence of scissural and/or hilar lymph node metastasis, in favorable topographical situations or in particular clinical situations (high operative risk (expected mortality for a lobectomy 5%), synchronous or metachronous multifocal tumors), an anatomical segmentectomy can be proposed [5,6]. In recent years, minimally invasive thoracic surgery has developed considerably, mainly in Western countries, with the successive arrival of video thoracoscopy (VATS) and more recently, robotic surgery (RATS). In 2007 and 2008, senior surgeons were less inclined to perform VATS than younger surgeons. After that, the minimally invasive approach gained popularity in countries including the USA and Spain, as a result of its efficacy and safety. Minimally invasive approaches are now recommended for early-stage localized lung cancer, as indicated in the ESMO Clinical Practice Guidelines [7], depending on the surgeon’s expertise, and provided that he or she is able to completely remove the tumor [5,6]. 

Several publications assessing quality of care [8,9] have been based on two quality indicators (e.g., 30-day mortality and failure-to-rescue). National administrative databases that provide data relative to all patients and all care centers nationwide are important tools for assessing quality of care. For example, the French national administrative database for hospital care (PMSI) provides a huge amount of epidemiological information about hospitalized French patients [10,11,12,13,14,15]. It includes a large national cohort of patients operated on for lung cancer (about 10,000 patients/year), with exhaustive recruitment in all French hospitals (137 in 2017–2018). 

However, one recurring critique, is that certain prognostic factors cannot be taken into account in national hospital databases [16,17,18]. For example, for lung cancer surgery, variables such as preoperative forced expiratory volume (FEV1), American Society of Anesthesiologists score (ASA score), body mass index (BMI), and TNM stage are not considered in studies based on these data [16,17,18]. 

On the other hand, these variables are present in clinical databases such as the Society of Thoracic Surgeons database or the French Epithor database (national database of thoracic surgery in lung cancer), which depends on the cooperation of 112 thoracic surgery centers [19,20,21]. Even if the demographic characteristics, risk factors, and outcomes in our previous study population [16] were very similar to those in previous French studies from the Epithor database [6], the absence of prognostic factors may call into question the validity of the various models based on a national hospital database. We used the same data from the Epithor clinical database as those used to validate the thoracoscore and which have been published [21].

To address these concerns, we created two logistic regression models, one developed with the French national hospital database and the other with the Epithor clinical database to analyze 30 day mortality. 

The main objective of this study was to estimate the performance of the two models based on the two different databases (hospital database and the Epithor clinical database) using different statistical criteria including calibration and discrimination. 

## 2. Materials and Methods

### 2.1. French Hospital Database

The French national hospital database (PMSI) was inspired by the US Medicare system. This database provides detailed medical information on all admissions to public and private hospitals in France, including discharge diagnoses, according to the tenth edition of the International Classification of Diseases (ICD-10) [22,23] and medical procedures coded according to the Common Classification of Medical Procedures (CCAM).

From this database, we included all patients with a pulmonary resection between 2005 and 2020. These patients were identified through a principal discharge diagnosis of lung cancer (ICD-10 code C34), associated with a procedure of lung cancer surgery (CCAM codes [24] of thoracotomy, video assisted thoracic surgery (VATS) or robot-assisted surgery) during the same hospital stay. For all patients, lung cancer was established by pathological analysis according to the World Health Organization’s 2004 classification of lung tumors [22].

#### 2.1.1. Patient Characteristics 

At baseline, we identified for all patients age and gender, but also surgery-related variables such as the surgical approach (thoracotomy, VATS, or robot-assisted surgery), the type of resection (limited resection, lobectomy, bi-lobectomy, and pneumonectomy), bronchoplasty, and the extent of the pulmonary resection. We also included comorbidities such as pulmonary disease (chronic bronchitis or emphysema), heart disease (coronary artery disease, cardiac arrhythmia, congestive heart failure, valvular heart disease, pulmonary artery hypertension, or pulmonary embolism), peripheral vascular disease, liver disease, cerebrovascular events, neurological diseases (hemiplegia or paraplegia), renal disease, hematologic disease (leukemia or lymphoma), metabolic disease (including obesity), anemia, other therapies (preoperative chemotherapy including neoadjuvant therapies, steroids) and infectious disease. Finally, we calculated a modified Charlson Comorbidity Index (CCI) as a marker of comorbidity [25].

#### 2.1.2. Hospital Characteristics 

In France, hospitals are classified as either non-academic public, academic public, private non-profit, or private for profit. For each hospital, we also calculated the hospital volume, defined as the median number of thoracic procedures performed per year. For the analysis, hospital volume was represented as a continuous variable that was transformed into a logarithm.

#### 2.1.3. Ethics

Patient consent was not required, seeing as the French national hospital data are based on pseudonymized data, i.e., they do not contain any identifying data. Consequently, patient-identifying information was not used. The patient’s identity is pseudonymized, which allows data from the same patient to be linked without knowing the patient’s identity. The study was conducted according to the guidelines of the Declaration of Helsinki, and approved by the National Committee for data protection: declaration of conformity to the methodology of reference 05 obtained on 7 August 2018 under the number 2204633 v0.

### 2.2. Clinical Database Epithor 

The database of the French society of thoracic and cardiovascular surgery, Epithor, was created in 2003 [26,27,28,29,30]. Currently, 112 centers use this database to store their data. Epithor underwent a significant transformation in 2016, and surgeons can now save patient data directly to a website called web Epithor. Several published articles have based their research on data extracted from the Epithor database [26,27,28,29,30]. As previously described, we included patients operated on for lung cancer from 1 January 2016 to 31 December 2018 and were entered in the Epithor database. 

#### 2.2.1. Patient Characteristics 

Baseline demographic data included sex, age, BMI, performance status, ASA score, FEV1, dyspnea score, and TNM stage. The number of comorbidities per patient were grouped into 4 values (0, 1, 2, and ≥3) [27]. The following surgery details were recorded as well: surgical approach (open thoracotomy and video-assisted thoracoscopy), type of surgery (wedge, lobectomy, bilobectomy, or pneumonectomy). 

#### 2.2.2. Hospital Characteristics 

In France, hospitals are classified as either non-academic public, academic public, private non-profit or private for profit. For each hospital, we also calculated the hospital volume defined as the median number of thoracic procedures performed per year. For the analysis, hospital volume was represented as a continuous variable that was transformed into a logarithm.

#### 2.2.3. Missing Data

The TNM stage data are not all complete and we thus created an additional class for missing data.

#### 2.2.4. Ethics

Use of this database was approved by the National Commission for Data Protection (CNIL No 809833) and this study adhered to the tenets of the declaration of Helsinki.

#### 2.2.5. Outcome Measurements

To assess the quality of care, we chose one outcome indicator identified at the patient level: 30-day mortality. In the calculation of 30-day in-hospital mortality, we included deaths during the surgical admission, as well as deaths that occurred during a subsequent hospital stay, within 30 days of the surgical admission.

#### 2.2.6. Statistical Analysis

First, we sampled the hospital database to obtain the same number of patients as the Epithor database over the same period (2016–2018).

Then, for each of the two databases, we randomly sampled a training dataset with 70% of the data and a validation data set with 30%.

We used a bootstrap backward procedure to determine which of these factors were significantly associated with the outcome in logistic regression models for the hospital database and the Epithor clinical database. Using this approach, 1000 replicated bootstrap samples were selected from the original data. Risk factors selected in at least 500 samples (50%) of the replicates were included in the model [31].

For continuous variables, we tested various extensions of the basic “linear predictor” models that can relax the linearity assumption, such as restricted cubic splines and fractional polynomials [32]. 

#### 2.2.7. Validation of Models

The performance of the models was assessed using the Brier score, Brier Max and Brier scaled [33].

The area under the receiver operating characteristic (AUC ROC) curve, concordance, and discrimination slope were used to measure the discriminatory ability of the model [33]. 

The calibration of the model was estimated by the relationship between the predicted probability and the observed outcome in that sample. Calibration by plotting predicted against observed probability can estimate intercepts and slopes of curves to quantify overfitting. Well-calibrated models have a slope of 1, whereas models that provide overly extreme predictions have a slope of less than 1; low predicted probabilities are too low, and high predicted probabilities are too high. 

The calibration of the model was assessed with the Hosmer–Lemeshow goodness-of-fit test [34]. We used the Integrated Calibrated Index (ICI), E50, E90, and Emax to quantify the calibration of logistic model regression [35]. The ICI can be interpreted as the weighted difference between observed and predicted probabilities, in which observations are weighted by the empirical density function of the predicted probabilities.

#### 2.2.8. Sensitivity Analysis

We carried out a sensitivity analysis on the Epithor database using only complete data.

The calculations for the logistics regression models were carried out using STATA 18 software (StataCorp, College Station, TX, USA), and R version 4.2.2 statistical software (http://www.r-project.org, accessed on 1 July 2023).

The statistical significance threshold was set at <0.05. However, in addition to variables of clinical interest, the variables with a *p*-value <0.20 in univariate analysis were included in the multivariate model.

## 3. Results

From the Epithor clinical database, 15,023 lung cancer surgery patients were analyzed, with the training data set comprising 10,516 patients and the validation data set comprising 4507 patients. There were 227 (2.16%) deaths in the training data and 93 (2%) deaths in the validation data. 

In parallel, 15,023 lung cancer surgery patients were identified in the hospital database, with 10,516 in the training data and 4507 in the validation data. There were 283 (2.7%) deaths in the training data and 119 (2.64%) deaths in the validation data.

### 3.1. Description of Predictors

For the Epithor clinical database and the hospital database, the description of patients and hospitals characteristics (type of hospital, hospital volume) is reported in the Appendix A. Variables such as FEV1, BMI, ASA score, performance status, and TNM stage can only be presented for the Epithor clinical database (Appendix A). Hospitals characteristics (type of hospital and hospital volume) were not significantly related to postoperative mortality; hospital volume was retained for the multivariate analysis (*p* < 0.20) in the hospital database (Appendix A Appendix A).

### 3.2. Training Model 

For the Epithor clinical database, 15 predictors were selected in >50% of bootstrap samples (Table 1). We used the restricted cubic spline function for the FEV variable, which was tested to make the model the most stable (Table 1). For the BMI variable, we performed a cubic transformation, as the linearity assumption was not met (Table 1). However, the linearity of the model was valid for age (Table 1).

For the hospital database, 15 predictors were selected in >50% of bootstrap samples (Table 2). For the age variable, we used the restricted spline function, and we transformed the hospital volume variable into a logarithm because the linearity assumption was not met and also to make the model more stable (Table 2). 

Age, sex, sleeve, extended resection, and VATS were the variables common to both databases and included in both models to explain 30-day mortality. The effects of these variables were in the same direction regardless of the model used. In particular, age, sleeve, and extended resection were positively associated with in-hospital mortality, unlike women and VATS.

### 3.3. Model Validity

The overall performance measures of the two models are reported in Table 3. The Brier score was identical in the training data and validation data for both databases (Table 3). The Brier score values were similar in the models of the two databases (Table 3). The estimated values of the Brier score for both models were far from 0.5, reflecting a non-informative model.

Discriminative ability was estimated by the AUC ROC curve, which was 0.83 (95% confidence interval (CI) 0.8–0.85) for training data from the hospital database and 0.8 (95% CI 0.76–0.84) for validation data (Table 3). For training data from the Epithor clinical database, the AUC ROC was 0.78 (95% CI 0.75–0.81) and for validation data it was 0.73 (95% CI 0.68–0.78) (Table 3). The model developed by the hospital database had a better discriminative value between living and deceased patients than the Epithor clinical database model, particularly for the validation data.

For the goodness-of-fit, we used the Hosmer–Lemeshow test, which was non-significant for both databases (Table 3). The calibration plot is shown in Figure 1 and Figure 2 for validation data (and in Appendix A for training data). For the validation data of the hospital database, the slope was less than 1 (Figure 1), so the high-predicted probabilities were too high. The calibration plot for validation data from the Epithor clinical database showed that the slope was even further away from 1, since the low predicted probabilities were too low, and the high predicted probabilities were too high (Figure 2).

The Integrated Calibration Index was comparable for both databases (Table 3), estimated at 0.0037 for validation data from the hospital database and 0.003 for validation data from the Epithor clinical database.

### 3.4. Sensitivity Analysis

Following the sensitivity analysis on the complete data on the Epithor database, we found that the results obtained in terms of model performance were not better (Appendix A).

## 4. Discussion

The model developed from the French national hospital database had nearly the same discriminative ability as the model developed from the Epithor clinical database. Overfitting was greater for validation data from the Epithor clinical database than from the hospital database, but the other measures of model performance were similar between the two.

This work showed that the performance of the models based on the PMSI hospital database was similar to that of the Epithor clinical database, using several statistical criteria (such as calibration and discrimination). However, this hospital database does not include variables with a major prognostic role, such as TNM stage, ASA score, FEV1, and BMI, which are taken into account by surgeons when they are making surgery-related decisions. 

One of the strengths of the Epithor clinical database is the presence of TNM stage, which is a variable that is recognized as being essential in prognostic/predictive models [27,28,29,30]. It is not clear why the performance of the two models was similar when the clinical variables used for the therapeutic indication are not included in the hospital data. In fact, the indication for surgery is based on the TNM stage: for example, surgeons will only be able to propose a minimally invasive approach (VATS or robot) for patients with a tumor classified as T1a or T1b. On the other hand, extended resection will be performed on patients classified as T2 or T3. However, it is possible that some variables describing surgery have a statistically significant effect in the hospital database model, partially compensating for the absence of the TNM stage. Variables describing the type of surgery, such as “extended resection” or “approach”, do appear to have a greater impact in the hospital database model than in the Epithor clinical database model.

Other clinical variables available in the Epithor clinical database, such as FEV1, dyspnea score, or gold score, are important to consider in a prognostic model as they reflect the patient’s pulmonary condition at the time of lung resection. However, even if they are not included as such in the PMSI database, the patient’s respiratory status is described globally by the variable pulmonary disease in the hospital database model, and this variable did have a significant effect in the model.

In the Epithor clinical database, the “performance status” variable is used to indicate the general condition of patients. Although this variable is not available directly in the hospital database, it can be taken into account indirectly in the model through other markers of comorbidities, such as the CCI. 

### Limitations

The first limitation of this study relates to the outcome chosen: some articles recommend a 90-day outcome rather than a 30-day outcome, which may be considered a major limitation of our work [36]. 

The quality of coding information in the hospital database can be questioned. For example, the risk that certain comorbidities might be underestimated cannot be ruled out. Indeed, coding practices may vary from one hospital to another, and involve different personnel (clinicians or technicians specialized in coding). Nevertheless, the quality of coding is verified by medical information professionals in each hospital (internal quality assessment). In France, the quality of comorbidity coding has increased significantly in recent years [37], particularly because of its impact on hospital funding. In addition, a national external quality assessment program has been set up to verify the quality of discharge abstracts in each hospital. All of these measures contribute to the quality of the data in this database, as our study seems to confirm. Other limitations of the PMSI administrative database must be emphasized, not only because of the missing variables of clinical interest, but also because of the lack of exhaustiveness of postoperative information, with difficulties in identifying unscheduled re-hospitalizations and the impossibility of knowing about events occurring at home in the first 90 days.

The quality of the Epithor clinical database used in this study may also be questionable, as suggested by a study conducted by our team in 2019 [38]. We also found that the TNM stage variable could present missing data, as shown by the calibration with a significant overfit for the validation data. We also underscored that not all French teams that perform lung cancer surgery contribute to the Epithor clinical database [38]. However, the Epithor database used in this study has become obsolete due to changes since 2016. In particular, deaths are no longer declared, but have been identified since 2020 by being matched with the INSEE death database.

An interesting means of overcoming the limitations of these two databases would be to link them. This would enable us to supplement French hospital data with clinical information available in Epithor for patients present in both databases. We can reasonably assume that patients present in the Epithor database are also present in the PMSI database. Hospital data are entered in a standardized and compulsory way for all hospitalized patients, which would therefore provide us with data for all patients who undergo lung cancer surgery nationwide.

## 5. Conclusions

In conclusion, this work shows that the performance of a model developed from the French national hospital database is nearly as good as the performance obtained with the Epithor clinical database, but that the hospital database lacks crucial clinical variables such as FEV1, ASA score, or TNM stage. To compensate for the absence of these variables of major interest, linking the French hospital data with Epithor data would be of interest. This linkage would make it possible to supplement standard clinical information with the additional relevant details found in Epithor.

## Figures and Tables

**Figure 1 cancers-16-00734-f001:**
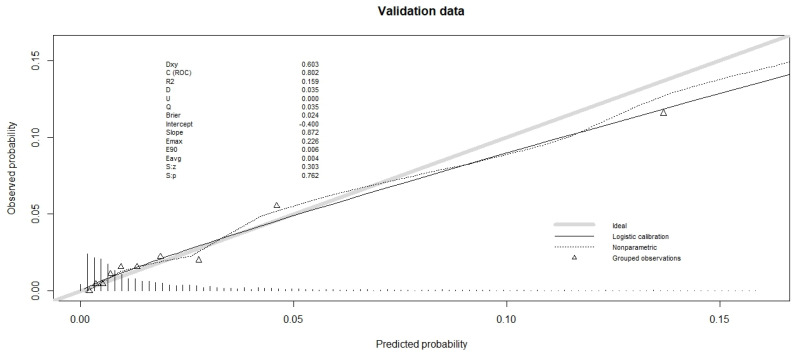
Calibration plot of observed mortality vs predicted mortality for validation data of hospital database (*n* = 4507).

**Figure 2 cancers-16-00734-f002:**
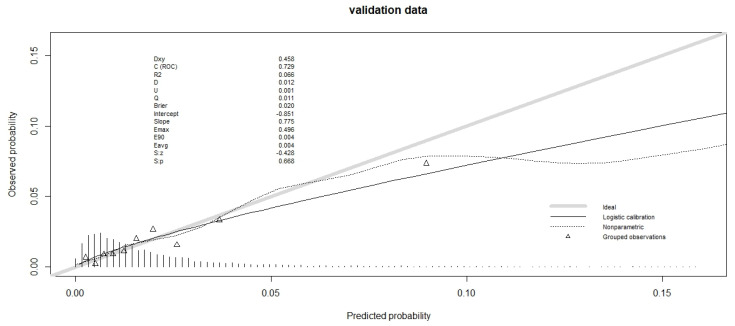
Calibration plot of observed mortality vs predicted mortality for validation data of Epithor clinical database (*n* = 4507).

**Table 1 cancers-16-00734-t001:** Logistic model regression developed with training data from the Epithor clinical database (*n* = 10,516).

	Coef	S.E.	Wald Test	*p*-Value
Intercept	−8.0726	1.7008	−4.75	<0.0001
FEV 1 *	0.0192	0.0134	1.43	0.1518
FEV 2	−0.0835	0.0416	−2.01	0.0445
FEV 3	0.3055	0.1609	1.90	0.0576
Age	0.0247	0.0080	3.09	0.0020
Body Mass Index **				
BMI: X/10	1.3608	0.8036	1.69	0.0904
BMI: X^3^	−0.0813	0.0374	−2.17	0.0298
Performance status (ref = 0)				
1	0.3381	0.1559	2.17	0.0301
≥2	0.7057	0.2269	3.11	0.0019
Dyspnea score ≥ 2	1.3531	0.2899	4.67	<0.0001
Gold score ≥ 3	0.2870	0.2128	1.35	0.1774
Pneumonectomy	0.3655	0.2362	1.55	0.1219
Sleeve	0.9130	0.3111	2.94	0.0033
VATS	−0.0331	0.1540	−0.22	0.8296
Extended resection	0.3369	0.3686	0.91	0.3606
TNM stage				
T (ref = T1)				
T2	0.1873	0.1898	0.99	0.3237
T3	0.6567	0.2056	3.19	0.0014
T4	0.8566	0.2560	3.35	0.0008
T missing	2.1008	0.8099	2.59	0.0095
N (ref = N0 N1)				
N2	0.3969	0.2089	1.90	0.0575
N missing	0.7002	0.1853	3.78	0.0002
M (ref = M0)				
M1	0.7117	0.2662	2.67	0.0075
M missing	−2.3664	0.8259	−2.87	0.0042
Female	−0.7042	0.1807	−3.90	<0.0001
Asa score (ref = 0–1)				
Asa score 2	0.3854	0.2779	1.39	0.1656
Asa score 3	0.6555	0.2822	2.32	0.0202
Comorbidity score ≥ 3	0.2693	0.1479	1.82	0.0687

* Restricted cubic splines. ** fractional polynomial.

**Table 2 cancers-16-00734-t002:** Logistic model regression developed with training data from the hospital database (*n* = 10,516).

	Coef	S.E.	Wald Test	*p*-Value
Intercept	−4.3107	1.0408	−4.14	<0.0001
Pulmonary disease	1.4882	0.1401	10.62	<0.0001
Heart disease	0.4115	0.1412	2.91	0.0036
Peripheral vascular disease	0.4095	0.1652	2.48	0.0131
Neurological disease	0.5063	0.2213	2.29	0.0221
Liver disease	1.9270	0.3037	6.35	<0.0001
Renal disease	0.7027	0.2410	2.92	0.0035
Metabolic disease	−0.3874	0.1867	−2.07	0.0380
Anemia	0.3951	0.1483	2.66	0.0077
Infectious disease	1.4135	0.3245	4.36	<0.0001
Other disease	0.4097	0.1292	3.17	0.0015
Extended resection	0.6063	0.1632	3.71	0.0002
Sleeve	0.8957	0.2966	3.02	0.0025
Female	−0.7435	0.1680	−4.43	<0.0001
VATS/robot	−0.6010	0.1590	−3.78	0.0002
Age 1 *	0.0054	0.0163	0.33	0.7379
Age 2	0.0405	0.0161	2.52	0.0117
Logarithm hospital volume	−0.2163	0.0661	−3.27	0.0011

* Restricted Cubic Splines.

**Table 3 cancers-16-00734-t003:** Evaluation of performance of logistic regression model.

	Hospital Database		Epithor Clinical Database	
	Training Data (*n* = 10,516)	Validation Data (*n* = 4507)	Training Data (*n* = 10,516)	Validation Data (*n* = 4507)
Performance measures				
Brier score	0.024	0.024	0.02	0.02
Brier max	0.026	0.026	0.021	0.02
Brier scaled	0.08	0.07	0.03	0.08
Discriminative ability				
AUC ROC	0.83 [0.80–0.85]	0.80 [0.76–0.84]	0.78 [0.75–0.81]	0.73 [0.68–0.78]
Concordance statistic		0.82		0.73
Discrimination slope		0.08		0.03
Calibration				
Hosmer–Lemeshow test (Χ^2^) (*p*-value)	8.7 (0.36)	8 (0.5)	10.4 (0.24)	9 (0.43)
ICI		0.0037		0.003
E50		0.003		0.002
E90		0.006		0.005
Emax		0.15		0.68
Abs Calibr. Error *		0.006		0.005
Unreliability *p*-value		0.2		0.05

ICI: Integrated Calibration Index. * Mean Absolute calibration error.

## Data Availability

Regarding the French hospital database, we are not allowed to transmit these data. PMSI data are available for researchers who meet the criteria for access to these French confidential data (this access is submitted to the approval of the National Committee for data protection) from the national agency for the management of hospi-talization (ATIH-Agence technique de l’information sur l’hospitalisation). Address: Agence tech-nique de l’information sur l’hospitalisation 117 boulevard Marius Vivier Merle-69329 Lyon Cedex 03. Regarding the Epithor database, raw data are available upon reasonable request.

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
