# Peer review of "Is the Validity of Logistic Regression Models Developed with a National Hospital Database Inferior to Models Developed from Clinical Databases to Analyze Surgical Lung Cancers?"

_cancers, 2024, doi:10.3390/cancers16040734_

Round 1
Reviewer 1 Report
Comments and Suggestions for Authors
This is an interesting manuscript which aim is to estimate the performance of two models based on the two databases: clinical & medico-administrative databases.
Materials and Methods
1. Outcome measurements
· Rows 155-157.Please better explain how the 30-day mortality is defined: are there any differences on patients who died before and after 30 days? (What is ment by “who died later during the same hospitalization”?).
2. Statistical analysis
· Are the populations of the two databases overlapping? (Are they similar, for example in terms of age, severity of the disease or other factors? Are patients in a database also present in the other one?).
· The significant p-value is not reported (it is commonly used a value equal to 0.05 or 0.1). Please add this information because it is necessary to define the statistical significance.
3. Validation of models
· Why is R2 used, instead of the adjusted R2 ? The adjusted R2 would be more appropriated.
Results
1. Description of predictors
· Row 202: “it showed the difference between the two databases”. Please rephrase this sentence, because the difference in the two databases (administrative and clinical) is not “proved” (for example no statistical tests are used to compare performances evaluated in both databases).
· Row 205: it is stated that “Hospital characteristics are not significantly related to post-operative mortality”, but in tables S1 and S2 (clinical database Epithor, supplementary material) for example the variables “gender” or “performance status” have a statistically significant p-value (<0.001). Otherwise define which variables are included in "hospital characteristics".
· Row 206: “Hospital volume is significantly related to mortality in the medico-administrative database”. See table S3: since the level of statistical significance is not reported, it is difficult to say if a p-value equal to 0.091 can be considered statistically significant or not.
2. Model validity
· Rows 232 - 235; maybe there is a typing error in the R2 value reported here or in Table 3. Is its corrected value equal to 13% or 12% for the development data of the Epithor clinical database?
· The R2 values found are not indicative of a good fit (< 20%), while Brier Scores are very close to 0, indicating a model with almost perfect performance. How do you explain that?
Discussion
· Row 290: “the other measures of model performance were not very different”. It is highly suggested to use a statistical test to better quantify the difference among the two values.
Tables and figures
Table 1
· Check categories of variables written in supplementary tables and reported in Table 1 (an example: the “performance status” variable has levels 0,1 and 2 or others? See also “dispnea score”, etc)
Figures 1 and 2
· Maybe there is a typing error in the two captions, because there are n = 10,516 patients in the two development databases.
Supplementary material:
· From which statistical test the reported p-values in the S1, S2, S3, S4 tables come? Please add this information.
Author Response
Reviewer 1
We thank the Reviewer for taking the time to review our work and for providing guidance to substantially improve the content and presentation of our manuscript. We have modified the article according to your requests. You will find every modification in the text using track changes, and the lines are noted in the answer for every point below. We hope we have met your requirements to improve this paper.
This is an interesting manuscript which aim is to estimate the performance of two models based on the two databases: clinical & hospital databases.
Materials and Methods
- Outcome measurements
- Rows 155-157.Please better explain how the 30-day mortality is defined: are there any differences on patients who died before and after 30 days? (What is ment by “who died later during the same hospitalization”?).
We agree with the Reviewer that the definition of 30-day hospital mortality was not clear. We reworded this definition as follows: “In the calculation of 30-day in-hospital mortality, we included deaths during the surgical admission, as well as deaths that occurred during a subsequent hospital stay, within 30 days of the surgical admission.” (lines 210-213)
- Statistical analysis
2.1· Are the populations of the two databases overlapping? (Are they similar, for example in terms of age, severity of the disease or other factors? Are patients in a database also present in the other one?).
We agree that the populations of the two databases are overlapping. In theory, patients from the Epithor database are included in the PMSI database because the latter contains all hospital admissions for all patients in France. However, in practice, only a sample of the PMSI database (constructed by random selection) was studied in order to have the same number of patients in both databases and therefore the same statistical power.
2.2· The significant p-value is not reported (it is commonly used a value equal to 0.05 or 0.1). Please add this information because it is necessary to define the statistical significance.
We agree with the Reviewer that it is necessary to define the statistical significance.
The statistical significance threshold was set at < 0.05. However, in addition to variables of clinical interest, the variables with a p-value < 0.20 in univariate analysis were included in the multivariate model.
We have added these points in the Methods section (lines 256-258).
- Validation of models
- Why is R2used, instead of the adjusted R2 ? The adjusted R2 would be more appropriated.
Following comments from other Reviewers who advised us to delete R² (“which is more commonly used in multiple regression than in logistic regression analysis”), we have removed this indicator for the validation of the two models.
Results
- Description of predictors
1.1· Row 202: “it showed the difference between the two databases”. Please rephrase this sentence, because the difference in the two databases (administrative and clinical) is not “proved” (for example no statistical tests are used to compare performances evaluated in both databases).
We fully agree that our wording was not appropriate and that the difference in the two databases (administrative and clinical) is not proved as no statistical tests are used to compare performances evaluated in both databases. Furthermore, our aim was not to compare the two databases but to study the performance of the two models. As a consequence, we have deleted this sentence. We have also reworded several sentences, in particular the discussion and the conclusion to take this remark into account, which we felt was very important.
1.2· Row 205: it is stated that “Hospital characteristics are not significantly related to post-operative mortality”, but in tables S1 and S2 (clinical database Epithor, supplementary material) for example the variables “gender” or “performance status” have a statistically significant p-value (<0.001). Otherwise define which variables are included in "hospital characteristics".
We apologize for this error. We actually forgot to present the variable in the method section, which has now been done (lines 165-170 concerning the hospital database and lines 196-201 for the Epithor database).
We have also added these details in the Results section (lines 273-274 and 278-279).
1.3· Row 206: “Hospital volume is significantly related to mortality in the hospital database”. See table S3: since the level of statistical significance is not reported, it is difficult to say if a p-value equal to 0.091 can be considered statistically significant or not.
We fully agree with the Reviewer that a p-value equal to 0.091 cannot be considered statistically significant. However, this is only the univariate result. As usual, we set a much wider threshold for selecting the variables to be included in the multivariate model, which explains why this variable was retained for the multivariate modelling.
In fact, as explained in the method section, lines 256-258, variables with a p-value < 0.20 in the univariate analysis were included in the multivariate model.
We fully understand that it was not easy to find this variable in the multivariate model. In fact, in the multivariate model, we included the hospital volume variable not as a continuous variable but after logarithmic transformation. Indeed, we transformed the hospital volume variable into a logarithm because the linearity assumption was not met and to make the model more stable as mentioned in the Methods and Results sections (lines 169-170, 200-201, 299-300). Finally, Table 2 shows that the effect of the hospital volume variable is highly significant in the multivariate modelling (Table 2, p = 0.0011).
We have therefore amended the sentence as follows: « Hospital characteristics (type of hospital and hospital volume) were not significantly related to postoperative mortality; hospital volume was retained for the multivariate analysis (p<0.20) in the hospital database (supplementary material S1, S2, S3, S4).» (lines 278-282)
- Model validity
2.1· Rows 232 - 235; maybe there is a typing error in the R2 value reported here or in Table 3. Is its corrected value equal to 13% or 12% for the development data of the Epithor clinical database?
We have indeed made a mistake and apologize for it.
However, we have removed all R2 values from this article (see reply to point 3 of Materials and Methods).
2.2· The R2 values found are not indicative of a good fit (< 20%), while Brier Scores are very close to 0, indicating a model with almost perfect performance. How do you explain that?
We fully agree with you that these two indicators do not provide the same information on model performance. As indicated in the response to point 3 of Materials and Methods, the R2 indicator has been removed.
Discussion
- Row 290: “the other measures of model performance were not very different”. It is highly suggested to use a statistical test to better quantify the difference among the two values.
We would like to thank the Reviewer for this very important comment. We are well aware of the value of a statistical comparison. Unfortunately, as the models are run on two different databases, it is not possible, to our knowledge, to use a statistical test to compare the performance of the models. This is why we have used the indicators usually recommended for evaluating the performance of these models.
Tables and figures
Table 1
- Check categories of variables written in supplementary tables and reported in Table 1 (an example: the “performance status” variable has levels 0,1 and 2 or others? See also “dispnea score”, etc)
We would like to thank the Reviewer for this remark, which enabled us to correct our mistakes. It is true that the categories presented in these various tables were not consistent. We have made the necessary corrections in the Tables.
Figures 1 and 2
- Maybe there is a typing error in the two captions, because there are n = 10,516 patients in the two development databases.
We would like to thank the Reviewer for this comment, which enabled us to correct our mistake.
Supplementary material:
- From which statistical test the reported p-values in the S1, S2, S3, S4 tables come? Please add this information.
Categorical variables were compared with a Chi-square test and quantitative variables were compared using a Student's t-test.
We have added this information in the legend of the Supplementary Tables.
Reviewer 2 Report
Comments and Suggestions for Authors
The authors address an interesting problem with respect to databases derived from the hospital management system and from the hospital information system. There is indeed salient information to be extracted from both systems. The paper focuses on 30-day mortality prediction in surgical lung cancer patients based on data collected in the French hospital patient summary system (PMSI) and on data collected in a clinical system (Epithor). How do the two prediction models perform with respect to each other?
I have several concerns about this work.
Major points
- There is a highly significant and clinically relevant death rate difference (>0.5%) between the two databases which cannot be ignored: 2.68% (PMSI) and 2.11% (Epithor). This introduces a bias in the comparison.
- There is no mention as to how authors handled missing values in both databases.
- Were the proportions of missing values the same in the training sample and validation sample for both systems?
- The use of a “missing variable”(e.g., T missing) in the construction of the risk prediction index (see Table 1) makes no sense. Analysis should be redone.
- Data in Table 3 tend to evidence that 30-day survival can be better predicted by PMSI than by Epithor (amazing)!
Minor points
- The title of the paper should include “surgical lung cancers” because the conclusions of the paper may not be extended (or generalized) to another disease.
- Avoid the term “medico-administrative” in English; why not “social security” or “hospital patient summary” database?
- Why use backward procedure? (justify)
- For validation, forget about R² and keep AUC-ROC, concordance, and slope. For calibration, keep ICI.
- Calibration In Figures 1 and 2, delete development data plot and keep only validation data on a single Figure for PMSI and Epithor
- Supplementary Tables S1-S4, replace column percentages by row percentages which makes more sense.
- In the same Tables S1-S4, use fewer decimal digits (e.g., age write 64.7 vs. 64.0) so that results become easier to read.
Other details
- Line 36: numbers of deaths 22761 and 10356 (over which period?)
- Line 38: 92-77% ?
- Line 65: any more recent figure?
- Line 73: STS undefined
- Lines 87-88: no real discussion of this objective in the paper.
- Line 160: specify “same period”
- Line 161: “training dataset” should be preferred to “development dataset”.
- Lines 168-170: unclear
- Lines 172: delete R² which is more commonly used in multiple regression than in logistic regression analysis
- Line 191: write Epithor
- Table 3: there is a shift upwards in the two columns below Hosmer-Lemeshow test
Comments on the Quality of English Language
Minor check
Author Response
Reviewer 2
We thank the Reviewer for taking the time to review our work and for providing guidance to substantially improve the content and presentation of our manuscript. We have modified the article according to your requests. You will find every modification in the text using track changes, and the lines are noted in the answer for every point below. We hope we have met your requirements to improve this paper.
The authors address an interesting problem with respect to databases derived from the hospital management system and from the hospital information system. There is indeed salient information to be extracted from both systems. The paper focuses on 30-day mortality prediction in surgical lung cancer patients based on data collected in the French hospital patient summary system (PMSI) and on data collected in a clinical system (Epithor). How do the two prediction models perform with respect to each other?
I have several concerns about this work.
Major points
- There is a highly significant and clinically relevant death rate difference (>0.5%) between the two databases which cannot be ignored: 2.68% (PMSI) and 2.11% (Epithor). This introduces a bias in the comparison.
We would like to thank the Reviewer for this very important comment. We fully agree with the Reviewer that the two databases are not comparable in terms of population and available variables. Indeed, the Epithor database does not cover all hospital admissions, but does include clinical variables of major importance (such as TNM or FEV1, for example). The PMSI database has the advantage of including all the relevant hospitalizations, but does not have these major clinical variables. Both databases are of great epidemiological interest. This is why we wanted to assess the performance of the models based on these two databases.
We are well aware that it would have been extremely interesting to have a statistical comparison. Unfortunately, as the models were launched on two different databases, it is not possible, to my knowledge, to use a statistical test to compare the performance of the models.
For this reason, we have not carried out a statistical comparison and have simply used the indicators usually recommended for assessing the performance of these models.
- There is no mention as to how authors handled missing values in both databases.
We thank the Reviewer for this remark.
For the Epithor database, the TNM stage data are not all completed and we thus created an additional class for missing data. We have modified the presentation of the method to explain how missing data are taken into account (lines 203-204 for the Epithor database).
For hospital data, we have no way of measuring the rate of missing data. Filling in a discharge abstract for each stay is compulsory. However, as mentioned in the Discussion section, the quality of coding information in each discharge abstract can be questioned. For example, the risk that certain comorbidities might be underestimated cannot be ruled out. Nevertheless, the quality of coding is verified by medical information professionals in each hospital (internal quality assessment). It has been shown that the quality of comorbidity coding has increased significantly in recent years in France, particularly because of its impact on hospital funding. In addition, a national external quality assessment program has been set up to verify the quality of discharge abstracts in each hospital.
- Were the proportions of missing values the same in the training sample and validation sample for both systems?
For the Epithor database, the proportions of missing values are the same in the training sample and the validation sample because they are drawn at random from the initial database: we observe in supplementary Tables S1 and S2 (missing data category) that the percentage of missing data is similar in both samples.
For hospital data, we have no way of measuring the rate of missing data, as mentioned above.
- The use of a “missing variable”(e.g., T missing) in the construction of the risk prediction index (see Table 1) makes no sense. Analysis should be redone.
We introduced this variable to follow some guidelines. Indeed, as recommended by F.E. Harrell. (Regression Modeling Strategies: With Applications to Linear Models, Logistic Regression, and Survival Analysis. Paris, France: Springer, 2001) and E.W. Steyerberg (Clinical Prediction Models: A Practical Approach to Development, Validation and Updating. New York: Springer, 2009), it is important to test whether the missing data are significantly related to the outcome. This was the case in our study, as shown in Table S1.
Furthermore, given the high number of missing data for the TNM (16% for the T, 16% for the N and 17% for the M), we felt it was important to include all the modalities of the missing data variable in the model, as removing the class of missing data for the TNM risked biasing the estimation of the model coefficients. In addition, the introduction of the missing data variable in the model enabled us to highlight an interaction between this variable and the other variables of interest (including TNM) in the analysis of the effect on mortality, which underlines the impact of missing data on this analysis.
Following your judicious remark and the existence of a statistically significant interaction, we re-run the models by removing the missing data from the Epithor database. The coefficients of the model without the missing data are very different from those estimated from the model with the missing data: for example, the difference was respectively 56% for T2, 20% for T3 and 20% for T4. The results obtained in terms of model performance are no better. In fact, model performance deteriorated, as shown by the area under the curve and the calibration for the validation data. We have added the corresponding results and Tables in the supplementary file (see Tables S5 and S6) and explained the methodology used in the Materials and Method section, as follows:
Methods (lines 249-251):
« Sensitivity analysis
We carried out a sensitivity analysis on the Epithor database using only complete data. »
Results (lines 365-368):
« Sensitivity analysis
Following the sensitivity analysis on the complete data on the Epithor database, we found that the results obtained in terms of model performance were no better (Tables S5 and S6). »
- Data in Table 3 tend to evidence that 30-day survival can be better predicted by PMSI than by Epithor (amazing)!
We agree that the results may seem surprising. As we stated above in response to your first point, the two databases cannot be compared. Consequently, it is not possible to draw any definite conclusions from these results. Moreover, even if the PMSI database has the advantage of including all relevant hospitalizations, it lacks clinical variables of major importance (such as TNM or FEV1, for example).
To help the reader better understand the interpretation that can be given to our results, we have modified the wording of the conclusion as follows (lines 453-465):
“In conclusion, this work shows that the performance of a model developed from the French national hospital database is nearly as good as the performance obtained with Epithor clinical database, but the hospital database lacks crucial clinical variables such as FEV1, ASA score or TNM stage. To compensate for the absence of these variables of major interest, linking the French hospital data with Epithor data would be of interest. This linkage would make it possible to supplement standard clinical information with the additional relevant details found in Epithor.”
Minor points
- The title of the paper should include “surgical lung cancers” because the conclusions of the paper may not be extended (or generalized) to another disease.
We fully agree with the Reviewer and have therefore added this term to the title, which is now: “Is the validity of logistic regression models developed with a national hospital database inferior to models developed from clinical databases to analyze surgical lung cancers?”
- Avoid the term “medico-administrative” in English; why not “social security” or “hospital patient summary” database?
We have replaced the term "medico-administrative database" with "hospital database" in the whole paper.
- Why use backward procedure? (justify)
We fully understand your comment. We have followed the recommendations of the authors stated previously (Harrell FE. Regression Modeling Strategies: With Applications to Linear Models, Logistic Regression, and Survival Analysis. Paris, France: Springer, 2001. Steyerberg EW. Clinical Prediction Models: A Practical Approach to Development, Validation and Updating. New York: Springer, 2009.) This choice is all the more recommended when the Bootstrap method is used to select the variables in the model.
- For validation, forget about R² and keep AUC-ROC, concordance, and slope. For calibration, keep ICI.
Thank you for your very pertinent suggestions. We have removed the R2 indicator.
- Calibration In Figures 1 and 2, delete development data plot and keep only validation data on a single Figure for PMSI and Epithor
We thank the Reviewer for this comment. We have replaced the initial Figures by the Figures related to the validation data. The Figures related to the training data have been moved in Supplementary Figures.
- Supplementary Tables S1-S4, replace column percentages by row percentages which makes more sense.
As requested by the Reviewer, we have replaced column percentages by row percentages.
- In the same Tables S1-S4, use fewer decimal digits (e.g., age write 64.7 vs. 64.0) so that results become easier to read.
As requested by the Reviewer, we have used fewer decimal digits.
Other details
- Line 36: numbers of deaths 22761 and 10356 (over which period?)
We would like to thank the Reviewer for his comments. We have specified in the text that we are referring to 2018 (line 71).
- Line 38: 92-77% ?
We have corrected by “77 to 92%” (line 74)
- Line 65: any more recent figure?
We would like to thank the Reviewer for his/her comments. We have updated the figures (lines 102-105). The number of hospitals performing lung resections for cancer has fallen, following regulations on authorizations. For 2017-2018, only 137 hospitals performed them for approximately 10,000 patients per year.
- Line 73: STS undefined
We have modified by “Society of Thoracic Surgeons” (lines 114-115)
- Lines 87-88: no real discussion of this objective in the paper.
We would like to thank the Reviewer for this very pertinent comment. We have in fact not discussed this point in this paper and have deleted this objective and the corresponding sentence from the introduction (lines 130-131).
- Line 160: specify “same period”
We have specified that this refers to the 2016-2018 period (line 219).
- Line 161: “training dataset” should be preferred to “development dataset”.
We have replaced the term “development dataset” by “training dataset” throughout.
- Lines 168-170: unclear
We would like to thank the Reviewer for this comment. We have reworded the sentence to clarify (lines 185-186).
- Lines 172: delete R² which is more commonly used in multiple regression than in logistic regression analysis
We fully agree. As suggested by the Reviewer, we have deleted R².
- Line 191: write Epithor
We have corrected accordingly (line 261).
- Table 3: there is a shift upwards in the two columns below Hosmer-Lemeshow test
We would like to thank the Reviewer and have made the appropriate corrections.
Round 2
Reviewer 2 Report
Comments and Suggestions for Authors
I have to acknowledge that the authors have responded appropriately to all my comments and remarks. They have to be commended for their excellent work.
The only ethical concern or question I have with this revised version of the paper relates to the authorship. Why do we have all of a sudden 5 more subjects in the authors' list? What were their respective contributions to the paper ? Personally, I feel this is contrary to scientific integrity ! The original authors should answer this question to the Editor.